# The Importance of Preconditioning for the Sonographic Assessment of Plantar Fascia Thickness and Shear Wave Velocity

**DOI:** 10.3390/s24144552

**Published:** 2024-07-14

**Authors:** Conor Costello, Panagiotis Chatzistergos, Helen Branthwaite, Nachiappan Chockalingam

**Affiliations:** 1Centre for Biomechanics and Rehabilitation Technologies, Staffordshire University, Stoke-on-Trent ST4 2DF, UK or conor.costello@tgh.nhs.uk (C.C.); h.r.branthwaite@staffs.ac.uk (H.B.); n.chockalingam@staffs.ac.uk (N.C.); 2Tameside and Glossop Integrated Care NHS Foundation Trust, MSK Podiatry Clinic, Ashton Primary Care Centre, 193 Old Street, Ashton-under-Lyne OL6 7SR, UK

**Keywords:** shear wave elastography, plantar fasciitis, plantar fasciopathy, plantar fascia, plantar aponeurosis, ultrasonography, diagnostic imaging, preconditioning, tissue stiffness, elasticity imaging techniques

## Abstract

Plantar fasciopathy is a very common musculoskeletal complaint that leads to reduced physical activity and undermines the quality of life of patients. It is associated with changes in plantar fascia structure and biomechanics which are most often observed between the tissue’s middle portion and the calcaneal insertion. Sonographic measurements of thickness and shear wave (SW) elastography are useful tools for detecting such changes and guide clinical decision making. However, their accuracy can be compromised by variability in the tissue’s loading history. This study investigates the effect of loading history on plantar fascia measurements to conclude whether mitigation measures are needed for more accurate diagnosis. The plantar fasciae of 29 healthy participants were imaged at baseline and after different clinically relevant loading scenarios. The average (±standard deviation) SW velocity was 6.5 m/s (±1.5 m/s) and it significantly increased with loading. Indicatively, five minutes walking increased SW velocity by 14% (95% CI: −1.192, −0.298, t(27), *p* = 0.005). Thickness between the calcaneal insertion and the middle of the plantar fascia did not change with the tissues’ loading history. These findings suggest that preconditioning protocols are crucial for accurate SW elastography assessments of plantar fasciae and have wider implications for the diagnosis and management of plantar fasciopathy.

## 1. Introduction

Plantar heel pain (PHP) is a common musculoskeletal complaint that leads to reduced physical activity and can have a detrimental effect on quality of life. It is among the most common foot conditions in older people and in athletic populations and it is estimated that one in ten people will develop PHP during their lifetime [1,2].

PHP can be caused by conditions that are local to the heel (e.g., plantar fasciopathy, heel fat pad atrophy) or systemic illnesses (e.g., seronegative spondyloarthropathies) [3]. Therefore, correctly identifying the underlying cause of PHP is critical for effective clinical management. Ultrasound imaging is a very useful tool to this end [4]. This is because the most common local causes of PHP are detectable in ultrasound and, compared to other medical imaging modalities, ultrasound is more portable, more cost-effective, and easier and safer to use [5].

Localized thickening of the plantar fascia above 4.0 mm is a diagnostic threshold for plantar fasciopathy, and hypoechoic regions in ultrasound imaging are indicative of fluid accumulation [5]. These features are associated with increased pain and hyperemia [6,7,8]. Even though sonographic findings can be reliable diagnostic markers, they do not seem to be robust measures for monitoring rehabilitation. Indeed, plantar fascia thickening appears to persist for many years despite improvements in symptoms [9]. Advances in ultrasound technology and the development of clinical shear wave (SW) elastography could help address this issue [10,11].

The plantar fascia is the most involved tissue in PHP. Its biomechanical role is to transmit forces from the rearfoot to the forefoot during locomotion [12]. To fulfil this role, the plantar fascia has a stiff fibrous structure that extends from the calcaneus and inserts at each of the metatarsophalangeal joint (MTPJ) capsules, with strips that continue to the proximal phalanges of each toe [13]. There are three parts to its structure—lateral, central, and medial—with the central part being the thickest [13]. The fibrous central portion is primarily constructed of type I collagen, arranged longitudinally, and as one moves distally from the proximal insertion it becomes thinner, fans out, and separates into five bundles attaching to each plantar plate of the MTPJs [6,7,13,14,15,16,17,18]. Type III collagen is more prevalent in areas where fibrous bundles are less well arranged, and loose connective tissue contains thin elastic fibers and hyaluronan [13]. Hyaluronan acts as a lubricant and shock absorber [19]. However, with increased water content, interstitial pressure is increased, and where focal accumulation occurs, gliding planes in fascia can be disrupted [19].

SW elastography measures the speed at which SWs propagate inside different tissues (in m/s), with increased or reduced SW velocity being indicative of increased or reduced tissue stiffness, respectively. It is important to note that due to the complex internal structure and biomechanics of musculoskeletal tissues, clinical SW elastography (in its current form) can only detect and quantify localized differences and (over-time) changes in stiffness, but not the absolute value of tissue stiffness [20,21]. Its capacity for quantitative assessment of differences or changes in tissue biomechanics is a key advantage of SW elastography over other clinical ultrasound elastography techniques (strain/real-time elastography [22,23]) that make it ideal for the study of overload injury and degenerative changes in musculoskeletal tissues [10,11,24,25]. Specifically in the case of the plantar fascia, SW elastography can detect degenerative changes associated with collagen breakdown, matrix degradation, increased mucoid content, and angiofibroblastic hyperplasia [10,11].

All the aforementioned degenerative changes lead to relative tissue softening. As a result, SW velocity in symptomatic populations is found to be significantly lower compared to their non-symptomatic counterparts [10,11,26,27,28,29,30]. In this case, increased SW velocity in the pathologic plantar fascia closer to normative values could also be a direct sign of recovery and a means to monitor the effectiveness of rehabilitation [31,32].

These findings highlight the potential value of SW elastography to complement current clinical practice for the diagnosis and rehabilitation monitoring of plantar fasciopathy. More research is needed to establish thresholds of normative or pathologic SW velocity in the plantar fascia and understand how these are affected by demographic, anthropometric, or biomechanical parameters. To this end, confounding parameters will have to be controlled to ensure that observed differences or changes in SW velocity are due to changes in tissue structure and biomechanics. Such confounding factors include imaging depth and direction, tissue thickness, and tissue loading history [33,34,35].

SW elastography measurements of the plantar fascia have been shown to be sensitive to long-term and to short-term effects of loading. More specifically, forefoot strike runners have been found to have reduced plantar fascia SW velocity compared to rearfoot strike runners [36], and habitual track runners tend to have increased SW velocity in their left foot compared to their right foot [37]. Regarding the immediate effects of loading, the literature indicates that plantar fascia stretching increases SW velocity [31], while running for 60 min reduces it [38]. Whilst some studies demonstrated an appreciation that loading history influences SW elastography measurements [36,37,38], its potential confounding effect (e.g., on baseline measures) was not considered.

In biomechanical research, the confounding effect of loading history is minimized with the help of preconditioning [39,40,41,42]. Preconditioning involves subjecting the tissue under study to a predefined loading scenario to ensure that loading history is always the same before any data are collected. However, to the authors’ knowledge, the effect of loading history on ultrasound measurements of the plantar fascia (thickness or SW velocity) has not been assessed and the literature is largely inconsistent regarding plantar fascia preconditioning.

In the minority of studies where the effect of loading history was considered, this was performed by maintaining the plantar fascia under constant loading for a predefined period before imaging. Usually this was minimal loading, and the participants were simply asked to lay still on the examination couch [8,43,44]. In one case, the plantar fascia was subjected to passive tension by keeping the toes in an extended position [45]. The duration of constant loading prior to imaging ranged from five minutes [43,44] to half an hour [8]. However, the effectiveness of these preconditioning protocols or indeed the need for them have not been tested.

Although important for mechanical characterization, the effects of loading history and preconditioning remain unexplored areas of research for studies involving plantar fascia thickness and SW velocity measurements. The current study therefore aims to establish if the loading history of the plantar fascia significantly influences these outcome measures, to determine whether a standardized preconditioning protocol is required in future clinical and research applications of measurements of plantar fascia thickness or SW velocity.

## 2. Materials and Methods

Twenty-nine healthy adults were recruited for this study (twelve female, seventeen male). Their average (±standard deviation) age and body mass index were 37 y (±12 y) and 26.2 kg/m^2^ (±3.8 kg/m^2^), respectively. People with a diagnosis of PHP in the last twelve months, a history of foot/ankle surgery, or diabetes were excluded. Pregnant women and people reporting pain in their feet on the day of testing were also excluded. Ethical approval was granted by Staffordshire University’s ethics committee (SU22-247_COSTELLO-RN). The participants provided informed consent before any data collection took place.

On entering the study, participants provided demographic information and were invited to remove shoes and socks to measure their body mass and height. Participants were asked to lay prone on the examination couch with feet hanging freely over the end of the couch (Figure 1a) [46].

Three imaging sites were defined based on the relevant literature [16] at the proximal, middle, and distal portions of the plantar fascia. The proximal imaging site was defined at the calcaneal insertion. The imaging site at the middle of the plantar fascia was adjacent to the navicular tuberosity. Finally, the distal site was immediately proximal to the 2nd metatarsal head. Before imaging the locations for the calcaneal insertion, the navicular tuberosity and the 2nd metatarsal head were identified with the help of ultrasound imaging and marked on the skin at the medial aspect of the foot to guide imaging (Figure 1a). Special care was given to ensure that minimal compression was applied during ultrasound imaging. The plantar fascia length was also measured with a ruler as the straight-line distance from the proximal insertion to the 2nd metatarsal head. All ultrasound measurements were taken using Aixplorer (SuperSonic Imagine, Aix-en-Provence, France) and a linear array probe (SuperLinear SL18-5 probe, “ankle/foot” preset). For all measurements, the probe was orientated in the same anterior to posterior direction.

Baseline ultrasound imaging was performed after five minutes of rest on the examination couch. To ensure that the baseline measurement was not time-dependent, the image capture process was repeated three times within a time-window of ≈2 min. Imaging was then repeated after five minutes [39] of treadmill walking at self-selected speed, after twenty star-jumps, and after twenty passive toe extensions. Participants were barefoot for each testing condition. The average (±standard deviation) speed and total distance during treadmill walking were 1.1 m/s (±0.2 m/s) and 328 m (±74 m), respectively. During passive toe extension, the participants remained in a prone position while the investigator passively extended all toes 20 times. One image per imaging site was analyzed for each loading scenario, bringing the total number of images captured for each participant to eighteen.

After the end of the testing session, thickness and SW velocity were assessed using the inbuilt software of the ultrasound unit. The thickness of the plantar fascia was measured using the digital caliper as a line connecting the tissue’s inferior border to the superior border at each of the measurement locations (Figure 1b–d). SW velocity (in m/s) was measured within a 1 mm^2^ circular area extending between the two boundaries of the plantar fascia and displayed as an average of all measurements within this area (Figure 2). At the end, thickness and SW velocity were recorded as the final outcome measures for each imaging site and load condition. Repeated measures ANOVA was used to ensure that SW measurements for the three recordings per condition were not time-dependent. Baseline results (SW and thickness) for each participant were calculated as the average of the measurements for these three images.

Microsoft Excel was used for descriptive statistics, and SPSS was used (version 28, IBM Corp., Armonk, NY, USA) for further statistical analysis. More specifically, data were screened for normality using the Shapiro–Wilk test. The association between SW velocity and thickness at the middle portion of the tissue was assessed for all loading scenarios using Spearman correlation analysis. Correlations between demographic or anthropometric parameters and baseline SW velocity and plantar fascia thickness were also investigated. The statistical significance of differences within and between loading conditions was assessed using repeated measures ANOVA and a paired-samples *t*-test, respectively (or their non-parametric equivalents).

Preliminary sample size calculations indicated that complete data sets for a minimum of 24 participants are needed to reliably assess the significance of medium effect size differences between conditions (two-tailed, effect size = 0.6 [47], a = 0.05, power = 0.80) [48].

## 3. Results

Plantar fascia thickness was successfully measured for all imaging sites and for all participants. SW velocity was successfully measured only for the middle portion of the plantar fascia. The ability to successfully image the proximal and distal portions in the majority of participants was confounded by bony structures [21]. As a result, SW measurements at these sites were excluded and the analysis of SW data will focus only on the middle portion of the plantar fascia.

All SW measurements were normally distributed and will be presented using their average (±standard deviation). However, some thickness measurements were not normally distributed. For consistency, all thickness measurements will be presented by their median (maximum value, minimum value).

Repeated measures ANOVA did not identify any statistically significant difference between the three baseline recordings of SW velocity, indicating that these measurements were not time-dependent. The same conclusion was also drawn for thickness measurements based on related-samples Wilcoxon signed rank tests.

Baseline SW velocity at the middle portion of the plantar fascia was equal to 6.5 m/s (±1.5 m/s), and it increased to 7.4 m/s (±1.3 m/s) after five minutes of walking (Figure 3). This was a statistically significant change of 14% (95% CI: −1.192, −0.298, t(27), *p* = 0.005, effect size = 0.60). SW velocity after 20 star-jumps was equal to 7.1 m/s (±1.0 m/s) and remained statistically significantly higher (by 9%) than baseline (95% CI: −1.132, −0.094, t(28), *p* = 0.022, effect size = 0.45). SW velocity after 20 passive extensions was 7.1 m/s (±1.4 m/s). This 9% difference from baseline was also statistically significant (95% CI: −1.085, −0.065, t(28), *p* = 0.029, effect size = 0.42).

On average, the plantar fasciae imaged here were 135 mm (±10 mm) long and their thickness gradually decreased from their proximal to distal portions. At baseline, median thickness decreased from 3.0 mm (5.1 mm, 2.5 mm) at the proximal portion to 1.7 mm (3.0 mm, 0.7 mm) at the middle portion and 1.5 mm (1.9 mm, 0.77 mm) at the distal portion of the plantar fascia. The related-samples Wilcoxon signed rank test indicated that these differences in thickness were statistically significant (*p* < 0.05).

Related-samples Wilcoxon signed rank analysis also indicated that thickness at the proximal or middle portions of the plantar fascia did not change significantly relative to baseline after walking, jumping, or passive toe extensions (Table 1). However, thickness at the distal portion statistically significantly reduced relative to baseline after walking (z = −3.383, *p* < 0.001). No other statistically significant difference was observed for thickness.

Spearman correlation analysis revealed a significant medium-strength association between SW velocity and thickness at the middle part of the plantar fascia after jumping (r = 0.613, N = 29, *p* < 0.001). In this case, increased SW velocity was associated with increased thickness (Figure 4). No other significant correlation was found between SW velocity at the middle part of the plantar fascia and thickness. Increased baseline SW velocity was strongly associated to increased plantar fascia length (r = 0.534, N = 29, *p* = 0.003) and weakly associated to increased participant height (r = 0.380, N = 29, *p* = 0.042).

Baseline thickness was significantly correlated to body mass for all three imaging sites, with higher body mass being associated with increased thickness. The correlations at the proximal (r = 0.603, N = 29, *p* < 0.001) and middle sites (r = 0.462, N = 29, *p* = 0.012) were of medium strength, while the correlation at the distal portion was weak (r = 0.374, N = 29, *p* = 0.046) [49]. 

## 4. Discussion

Biomechanical characterization of human tissues plays a crucial role in the study of tissue injury and rehabilitation. In the context of this research, ultrasound assessment of plantar fascia SW velocity and thickness can significantly contribute to the effective management of PHP by facilitating the diagnosis of the underlying pathology and guiding clinical decision making. In biomechanical research, it is established that such sonographic evaluations are likely to be affected by the tissue’s loading history, and appropriate preconditioning protocols are put in place to enhance their accuracy, reliability, and relevance [39,40,41,42]. This study is the first to establish whether standardized preconditioning protocols are also needed for the sonographic examinations of the plantar fascia used in the clinical management of PHP.

The results discussed within this study clearly indicate that measurements of plantar fascia thickness at the proximal and middle parts of the tissue are not significantly affected by loading history. The proximal and middle parts of the plantar fascia are also the areas where plantar fasciopathy, a common cause of PHP, is mostly detected [1,6]. Only thickness at the distal portion of the tissue close to the second metatarsal seemed to significantly decrease relative to baseline after walking. This portion of the plantar fascia was included in the analysis for completeness. However, pathologies detected there are rarely linked to PHP. Based on this, it can be concluded that the results of this study do not support the need for any preconditioning for the measurement of plantar fascia thickness for the diagnosis of plantar fasciopathy.

At this point, it is very difficult to explain why thickness at the distal end of the plantar fascia appears to be more sensitive to loading history than the tissue’s middle and proximal parts. This finding could be indicative of a non-uniformity in material properties and/or in the distribution of loading [14,45]. Computational research has also highlighted the area of the second metatarsal head as an area of intense concentration of mechanical stress [50]. Further research and independent confirmation are needed before any conclusions are drawn regarding possible differences between the distal, middle, and proximal parts of the plantar fascia.

Contrary to thickness, measurements of SW velocity appeared to be significantly affected by loading history. In this case, SW velocity significantly increased after walking, and this difference remained statistically significant when jumping or passive extensions were added to the tissues’ loading history. The increase in SW velocity due to the effect of loading history was between 9% and 14%. The observed magnitude of change is clinically relevant and could have a detrimental effect on clinical decision making [10,26].

In a recent study by Baur et al., the average SW velocities in the plantar fascia of people with plantar fasciopathy and their healthy counterparts were 4.98 m/s and 6.94 m/s, respectively [26]. Using a cutoff threshold of 6.16 m/s, Baur et al. were able to correctly classify healthy and pathologic tissues with a specificity of 81% and sensitivity of 79% [26]. Adding a potential error of ≈9–14% due to potential differences in the tissue’s loading history would significantly undermine the accuracy of this classification.

Based on these results, it can be concluded that SW elastography assessments of the plantar fascia for research or clinical application related to PHP must account for the potentially confounding effect of loading history. In this case, the use of a standardized preconditioning protocol is strongly advised to enhance the reliability of future research and clinical applications of plantar fascia SW elastography.

This preconditioning protocol could be simply to rest the tissue for a predefined amount of time [8,43,44]. Due to the inherent viscoelastic nature of the tissues of the foot, it is fair to assume that when all external loads are removed the plantar fascia will gradually revert to its baseline stiffness [51,52]. What is unknown is how much resting time is needed until a stable baseline stiffness is reached. In this study, and based on the previous literature, a resting period of five minutes was used prior to baseline imaging [43,44]. Three images per site were captured to confirm that the properties of the tissue did not undergo drastic changes over time. The first image was captured as close to the five-minute mark as possible. Because of the uniqueness of each participant, it was difficult to strictly control the exact time when each image was taken. We estimate that the time difference between images was ≈1 min. We hypothesized that if tissue properties had not stabilized, then a consistent pattern of change would become apparent between measurements. Repeated measures analyses indicated that this was not the case. Indeed, thickness and SW velocity values remained stable between recordings. This observation aligns with previous ex vivo results from the literature [51] and offers an initial support to the use of a five-minute rest protocol to minimize the effect of loading history on plantar fascia biomechanics. However, changes in plantar fascia biomechanics will have to be observed for longer periods of rest before concluding on the most appropriate preconditioning protocol [38]. Future research on this topic will also need to account for individual variability in tissue recovery across different loading scenarios.

This study highlights the sensitivity of the plantar fascia to loading history to draw attention to its potential confounding effect on results. At the same time, it is important to point out that the capacity of connective tissues to change their stiffness in response to mechanical stimuli is not an artifact but a tissue property in its own right [53]. Developing structured reliable protocols to quantify the effect of conditioning could open a new dimension in our understanding of tissue biomechanics, with significant potential applications for research and clinical practice.

SW velocity increased relative to baseline after walking, but remained relatively stable when jumping and passive extensions were added to the tissues’ loading history. Because the thickness of the tissue remained the same between all conditions, it can be concluded that the plantar fascia became stiffer after loading. 

Accounting for possible changes in plantar fascia thickness is necessary because of the confounding effect of guided wave propagation [34,35]. Guided wave propagation affects SW elastography in tissues which, like the plantar fascia, are relatively stiff and have a thickness smaller than the SW wavelengths [35]. In this case, the generated SWs are repeatedly reflected at the boundaries of the tissue, leading to a guided wave propagation along its length. This phenomenon is sensitive to thickness, which means that changes in thickness will directly change SW velocity even if tissue stiffness remains constant. Indeed, it has been demonstrated that guided wave propagation leads to increased SW velocity with increasing thickness [34]. This relationship between thickness and SW velocity was also observed here (Figure 4). This finding further highlights the importance of measuring plantar fascia thickness to infer changes or differences in plantar fascia stiffness based on measurements of SW velocity. 

Among the key limitation of this study was that recruitment was restricted to a healthy population. As a result, the measurements presented here cannot be used to predict the magnitude of change in SW velocity and thickness that loading can have in people with PHP. Even though the absolute values of the results are not transferable to pathologic populations, the conclusion that loading history can be a significant confounding factor still stands. Moreover, further research will be needed to conclusively identify the most appropriate preconditioning protocol for robust use of SW elastography in plantar fascia research and the management of PHP. 

At this point, it should be noted that in the absence of a validated preconditioning method it is not possible to isolate the effects of individual loading scenarios. This means that the respective measurements should be interpreted as assessments of the cumulative effect of all loading scenarios that preceded them.

## 5. Conclusions

Measurements of plantar fascia SW velocity appear to be significantly affected by the loading history of the tissue. Research studies and clinical protocols of SW elastography should include a standardized preconditioning protocol to minimize the effect of this confounding factor. These applications include but are not limited to the diagnosis and management of plantar fasciopathy.

On the contrary, measurements of plantar fascia thickness in the parts of the tissue that are most affected by plantar fasciopathy (proximal and middle) do not appear to be sensitive to loading history.

Because the thickness of the middle part of the plantar fascia remained unaltered, the observed differences in SW velocity can be interpreted as differences in tissue stiffness. More specifically, the results presented here indicate that five minutes of walking significantly increase stiffness in the middle part of the plantar fascia relative to baseline. This difference remained significant after jumping and passive extensions were added to the tissues’ loading history. Further research is needed to draw generalizable conclusions about the effects of different types, durations, intensities, and combinations of loading scenarios.

Overall, SW elastography can be a very useful tool for the diagnosis of conditions linked to altered tissue stiffness such as plantar fasciopathy. To this end, measurements of SW velocity should be complemented by measurements of tissue thickness to account for the confounding effect of guided wave propagation.

## Figures and Tables

**Figure 1 sensors-24-04552-f001:**
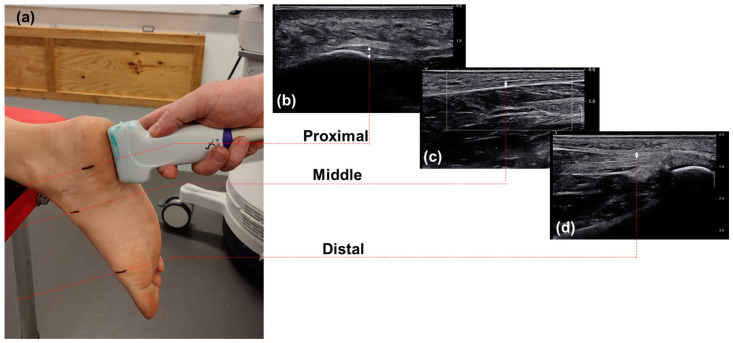
The imaging set-up (**a**) and typical ultrasound images for the measurement of plantar fascia thickness at the proximal (**b**), middle (**c**), and distal parts (**d**) of the tissue.

**Figure 2 sensors-24-04552-f002:**
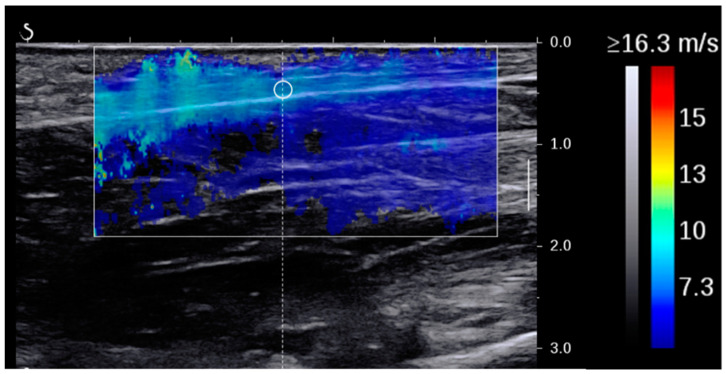
The measurement of shear wave velocity at the middle portion of the plantar fascia.

**Figure 3 sensors-24-04552-f003:**
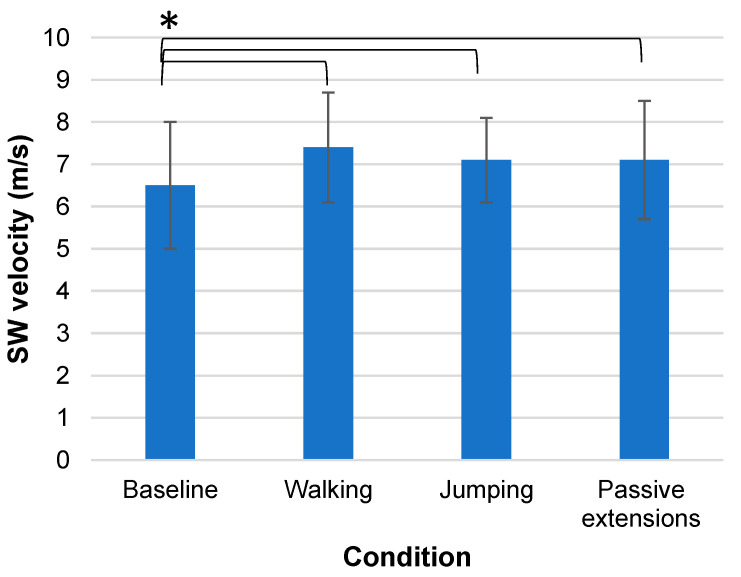
Population average SW velocities at baseline and after walking, jumping, and passive extensions. Statistically significant differences from baseline (*p* < 0.05) are indicated with (*) (paired-samples *t*-test).

**Figure 4 sensors-24-04552-f004:**
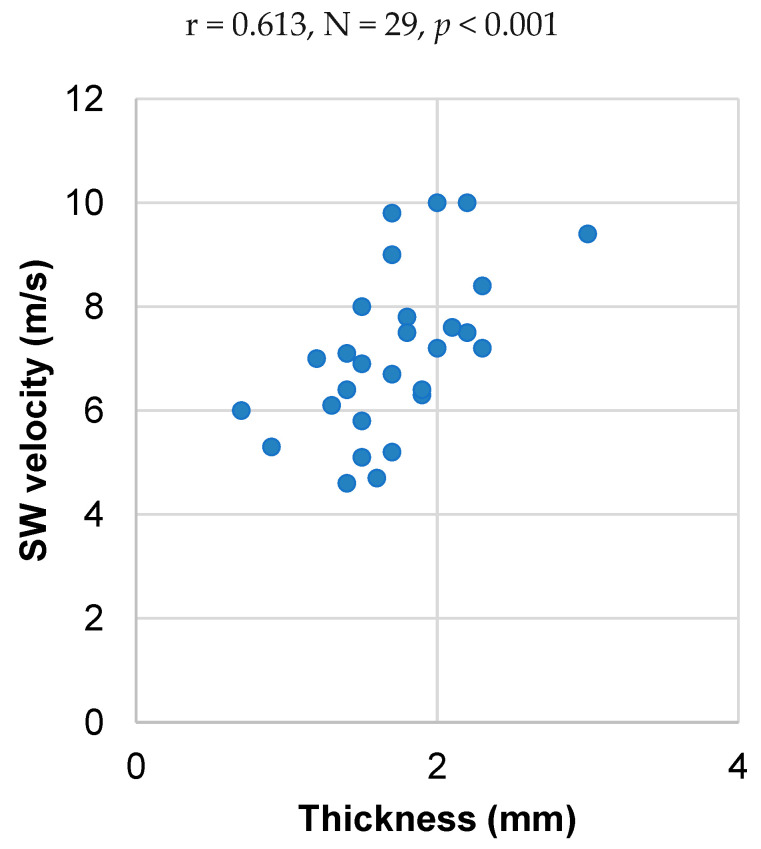
Statistically significant association between shear wave (SW) velocity and thickness in the middle part of the plantar fascia. The results presented in this graph were captured immediately after the participants had completed 20 star-jumps.

**Table 1 sensors-24-04552-t001:** Measurements of thickness (mm) in different parts of the plantar fascia (proximal, middle, distal) at baseline and after different load conditions reported as median (max, min). Statistically significant differences from baseline are indicated with (*).

		Imaging Site		
		Proximal	Middle	Distal
**Condition**	**Baseline**	3.0 (5.1, 2.5)	1.7 (3.0, 0.7)	1.5 (1.9, 0.8)
	**Walking**	3.2 (5.4, 2.4)	1.7 (2.2,1.1)	1.3 (1.8, 0.8) *
	**Jumping**	3.2 (4.7, 2.1)	1.7 (3.0, 0.7)	1.5 (2.2, 0.9)
	**Passive extensions**	3.3 (5.0, 2.1)	1.7 (2.7, 1.2)	1.4 (1.9, 0.8)

## Data Availability

The data that support the findings of this study are available from the corresponding author upon reasonable request.

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
