# Peer review of "The Importance of Preconditioning for the Sonographic Assessment of Plantar Fascia Thickness and Shear Wave Velocity"

_sensors, 2024, doi:10.3390/s24144552_

Round 1

Reviewer 1 Report

Comments and Suggestions for Authors

Please see my comments in the attached document.

Author Response

We would like to thank the reviewer for their constructive feedback. The manuscript is now revised according to their suggestions. Please see the attached document for a point-to-point discussion of individual comments and of the specific actions undertaken to address them.

Reviewer 2 Report

Comments and Suggestions for Authors

The study found that shear wave velocity in the plantar fascia increases significantly with loading, suggesting that preconditioning is necessary for accurate elastography imaging. However, the thickness of the plantar fascia did not change with the tissue's loading history. Overall, this study provides novel information, but the results should be presented more clearly.

 - Line 3: Please use “shear wave velocity” instead of “shear wave speed.”

- Line 153: Please also show the shear wave velocity measurements at distal and proximal sites.

- Lines 181-187: I think this paragraph contains the most important finding in your study. Please consider inserting a table or a figure here to make it clearer.

- Line 207: “No other significant correlation was found between shear wave velocity and thickness.” Please specify “other” clearly. Do you mean “proximal and distal”? Please also indicate “r=0.613, N=29, p<0.001” in Figure 3.

- Line 210: For Figure 3, showing only the middle part is not enough. Please consider using Figure 3(A), 3(B), and 3(C), etc., to more easily visualize and compare the association between shear wave velocity and thickness at different sites.

- Lines 310-311: “Measurements of plantar fascia shear wave velocity are significantly affected by the loading history of the tissue.” Please specify whether shear wave velocity increases or decreases, and at which sites it is affected.

Comments on the Quality of English Language

Good

Author Response

(The authors gave the same response as above.)

Reviewer 3 Report

Comments and Suggestions for Authors

Author Response

(The authors gave the same response as above.)

Round 2

Reviewer 2 Report

Comments and Suggestions for Authors

I appreciate the authors’ efforts in revising the manuscript. I have several minor comments and hope these comments may improve the manuscript further.

 1. In lines 47-62, the text is too lengthy. Please combine them into one paragraph and reduce the introduction related to MTPJ and hyaluronan. Similarly, in lines 81-98, these two paragraphs can be combined into one.

2. In the newly added Figure 3, please specify the statistical method used in the figure legend.

3. I agree with the conclusion that “Research studies and clinical protocols of SW elastography should include a standardized preconditioning protocol to minimize the effect of this confounding factor.” However, I don’t understand why “These results do not allow us to draw generalizable conclusions about the direction of the effect of loading history.” For example, in the newly added Figure 3, it seems that walking, jumping, and passive extensions significantly increase the SW velocity in the middle part of the plantar fascia. Why not mention this in the conclusion?

4. Please check the accuracy of your references. For instance, in line 71, “strain/real-time elastography [22]” should more appropriately cite “DOI:10.1148/radiol.11101665” and “DOI:10.1148/radiol.12120969” instead of [22], because you are discussing heel pain rather than ulceration. Additionally, in lines 95-96, reference [45] seems not relevant to your statement. Should it refer to [49]?

Author Response

We would like to thank the reviewer for their helpful feedback. The manuscript is now revised according to their suggestions. Please see the attached response to individual comments. New changes are highlighted green in latest version of the revised manuscript.
